# Exploring Conditions for Diffusion models in Robotic Control

## Abstract

While pre-trained visual representations have significantly advanced imitation learning, they are often task-agnostic as they remain frozen during policy learning. In this work, we explore leveraging pre-trained text-to-image diffusion models to obtain task-adaptive visual representations for robotic control, without fine-tuning the model itself. However, we find that naively applying textual conditions—a successful strategy in other vision domains—yields minimal or even negative gains in control tasks. We attribute this to the domain gap between the diffusion model's training data and robotic control environments, leading us to argue for conditions that consider the specific, dynamic visual information required for control. To this end, we propose **CoRoCo**, which introduces learnable *task* prompts that adapt to the control environment and *visual* prompts that capture fine-grained, frame-specific details. Through facilitating task-adaptive representations with our newly devised conditions, our approach achieves state-of-the-art performance on various robotic control benchmarks, significantly surpassing prior methods.

## 1 Introduction

Recent advances in diffusion models (Ho et al., 2020) have not only facilitated high-quality image synthesis, but also demonstrated as a strong visual representation for various vision tasks (Baranchuk et al., 2021). Among them, pre-trained text-to-image diffusion models, *e.g.* Stable Diffusion (Rombach et al., 2022), have shown that utilizing text *conditions* can significantly boost the performance in visual perception tasks, without the need for fine-tuning the model (Zhao et al., 2023). The key to leveraging text conditions lies in obtaining well-designed prompts (Kondapaneni et al., 2024)—often describing objects in the image or the given task—that can funnel useful information into downstream tasks. This not only enhances the proficiency of diffusion models on downstream tasks but also broadens their applicability to a wider variety of vision tasks (Yin et al., 2025; Wu et al., 2025).

Robotic control, meanwhile, has also benefited greatly with the introduction of pre-trained visual representations to imitation learning (Parisi et al., 2022). By leveraging frozen visual encoders pre-trained on large-scale data, these representations have replaced the previous *tabula-rasa* paradigm of training vision encoders from scratch on limited-scale control data. However, this approach is limited by its **task-agnostic** nature, as the visual representations remain frozen during downstream policy learning. Since the suitability of a representation for a specific task is unknown beforehand, determining which representation performs best often requires manual, task-by-task inspection (Majumdar et al., 2023), which becomes cumbersome given the vast variety of control tasks. While a straightforward solution might be to fine-tune the vision encoder, this often results in poor results as the model loses generalization capabilities by overfitting to specific scenes in imitation learning (Majumdar et al., 2023).

In this work, we explore bridging text-to-image diffusion models to robotic control for achieving **task-adaptive** visual representations through **conditions**, without fine-tuning the diffusion model. Inspired from the effectiveness of conditions in visual perception tasks, we ask following question: *How can we effectively implement conditions for diffusion models in robotic control?* We begin by investigating textual conditions, generating captions with a state-of-the-art vision-language model (Comanici et al., 2025) to observe their impact on control task performance. However, as shown in Fig. 1, the gains are minimal, and in some cases, performance even declines. This result contrasts sharply with findings

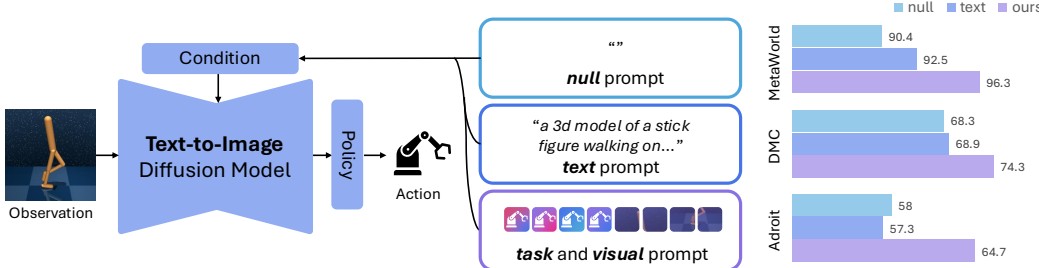

Figure 1: **How can we condition diffusion models in robotic control?** We investigate methods for *conditioning* text-to-image diffusion models (Rombach et al., 2022) to perform control, aiming to address various tasks in a *task-adaptive* manner. We observe that text prompts, unlike in other vision tasks (Zhao et al., 2023), are ineffective for robotic control. Therefore, we propose to learn **task** prompts in control environments and further incorporate dynamic details through **visual** prompts for conditioning diffusion models.

in other vision tasks (Zhao et al., 2023), where machine-generated captions have served as strong conditions (Kondapaneni et al., 2024).

Upon investigation, we find that pre-trained diffusion models often struggle to accurately associate text condition to the image in control environments. We attribute this discrepancy to the nature of the diffusion model being training on web-collected images, which suits visual tasks that involve real-world images and common objects, such as semantic segmentation. However, control environments, featuring specialized robotic agents performing specific tasks, would require a more careful and deliberate approach to devising effective conditions for downstream policy learning.

Robotic control tasks, unlike semantic segmentation Zhao et al. (2023); Kondapaneni et al. (2024), operate on dynamic video streams and require a finer visual granularity to interact with specific parts of objects, not just to categorize them. This dynamic nature implies that effective conditions must be generated uniquely for each frame (Hong et al., 2024) to guide evolving actions and adapt to changing visual states. Consequently, we hypothesize that conditions for control should incorporate **visual** information from every frame to capture both dynamic behavior and fine-grained details.

To this end, we propose a simple, yet effective method that incorporates visual information while addressing the limitations of text conditions. Specifically, we replace the text prompt with learnable **task** prompts, which are learned during downstream control tasks to ensure accurate grounding within the specific environment. Furthermore, to enable the conditions to capture the detailed visual state of each frame, we employ a vision encoder and utilize its representations as **visual** prompts. We demonstrate that both the task and visual prompts can be learned end-to-end during downstream policy learning using a standard behavior cloning objective.

Our framework for leveraging diffusion models with **co**nditions in **ro**botic **co**ntrol, **CoRoCo**, achieves state-of-the-art performance in robotic control tasks (Tassa et al., 2018; Yu et al., 2020; Rajeswaran et al., 2018), surpassing VC-1 (Majumdar et al., 2023). We verify our design choices by comparing to baselines with text conditions and different conditioning methods (Zhou et al., 2022; Kondapaneni et al., 2024) from visual perception tasks. In addition, we provide detailed analysis and ablations on our approach, highlighting the importance of conditions in diffusion models for robotic control.

## 2 RELATED WORK

### 2.1 PRE-TRAINED VISUAL REPRESENTATIONS FOR ROBOTIC CONTROL

In recent years, visual representations derived from self-supervised pre-trained models (Radford et al., 2021; Cherti et al., 2023; Majumdar et al., 2023; He et al., 2022; Caron et al., 2021) have demonstrated notable effectiveness in visuo-motor manipulation tasks (Parisi et al., 2022). Specifically, Parisi et al. (Parisi et al., 2022) showed that visual representations from frozen pre-trained encoders, such as MoCo (He et al., 2020) and CLIP (Radford et al., 2021), can not only outperform representations trained from scratch but are also comparable to ground-truth state features in behavior cloning. This finding has spurred extensive exploration into pre-trained visual representations for control, alongside

a search for self-supervised learning frameworks particularly suited for robotic manipulation. Among these, R3M (Nair et al., 2022) employs a time-contrastive learning objective on ego-centric data with vision-language alignment, whereas VIP (Ma et al., 2022) introduces value-implicit learning to associate goal and initial states. MVP (Radosavovic et al., 2023) and VC-1 (Majumdar et al., 2023) both adopt MAE (He et al., 2022) pre-training methodologies, curating large datasets that include ego-centric and instructional videos to enhance transferability to robotic manipulation tasks. More recently, SCR (Gupta et al., 2024) has investigated representations from Stable Diffusion (Rombach et al., 2022) for navigation and control tasks. Nonetheless, these methods opted for keeping the visual representation frozen, resulting them to be task-agnostic.

## 2.2 DIFFUSION MODELS AS PRE-TRAINED VISUAL REPRESENTATIONS

Recent advancements in diffusion models (Ho et al., 2020; Rombach et al., 2022) have enabled the synthesis of high-resolution images with unprecedented fidelity. This progress has concurrently motivated diverse investigations into the internal representations of generative diffusion models (Tang et al., 2023; Luo et al., 2023; Baranchuk et al., 2021; Zhao et al., 2023; Xiang et al., 2023) for various downstream vision tasks. DDPMSeg (Baranchuk et al., 2021) was among the first to explore the efficacy of diffusion model's representations in label-scarce segmentation, while DDAE (Xiang et al., 2023) focused on image classification. DIFT (Tang et al., 2023), DHF (Luo et al., 2023) and SD-DINO (Zhang et al., 2023) have demonstrated that the representation from diffusion models can achieve state-of-the-art in semantic correspondence tasks. Notably, VPD (Zhao et al., 2023) demonstrated that downstream performance can be enhanced by with text conditions, such as the names of objects present in an image, in tasks such as semantic segmentation and monocular depth estimation. SD4Match (Li et al., 2024) and EcoDepth (Patni et al., 2024) proposed prompting modules to derive conditions for semantic correspondence and monocular depth estimation. TADP (Kondapaneni et al., 2024) demonstrated that text descriptions generated from vision-language models can serve as strong conditions, and could be further enhanced with style modifiers learned from Textual Inversion (Gal et al., 2022). However, we distinguish our approach by focusing on robotic control, rather than for visual tasks in general image domains.

## 3 PRELIMINARIES

**Diffusion models** (Sohl-Dickstein et al., 2015; Ho et al., 2020; Kingma et al., 2021) constitute a class of generative models that learn to reverse a multi-step noising process, thereby reconstructing a target data distribution. In this work, we focus on conditional diffusion models (*e.g.* Stable Diffusion (Rombach et al., 2022)), which enable image generation guided by a condition $\mathcal{C}$, often being text prompts. The training objective is to reverse the noising process, typically discretized into $T$ timesteps. A pre-defined noise schedule, denoted by $\alpha_t$, facilitates the definition of the noised latent variable $z_t$ at timestep $t$ as:

$$z_t = \sqrt{\bar{\alpha}_t} z_0 + \sqrt{1 - \bar{\alpha}_t} \epsilon, \tag{1}$$

where $z_0$ is the initial clean data, $\bar{\alpha}_t = \prod_{i=1}^{t} \alpha_i$, and $\epsilon \sim \mathcal{N}(0, I)$ is Gaussian noise. Following Ho et al. (Ho et al., 2020), with appropriate parameterization, diffusion models can be trained by regressing the added noise $\epsilon$ from $z_t$:

$$\mathcal{L}_{\text{DM}} = \mathbb{E}_{z_0, \epsilon, t} \left[ \| \epsilon - \epsilon_\theta(z_t(z_0, \epsilon), t; \mathcal{C}) \|_2^2 \right], \tag{2}$$

where $\epsilon_\theta$ indicates the denoising network, typically a U-Net (Ronneberger et al., 2015) or a Transformer (Vaswani et al., 2017) architecture. Stable Diffusion, for our case, is a Latent Diffusion Model (LDM) (Rombach et al., 2022) with an U-Net architecture, in which the diffusion process occurs in a compressed latent space learned by an autoencoder, specifically a VQGAN (Esser et al., 2021). For conditional generation, U-Net-based LDMs implement Transformer blocks with cross-attention layers into the U-Net blocks to inject the condition $\mathcal{C}$ into the image generation process.

**Extracting visual representation from diffusion models.** To extract visual representations, initially, an input image $I$ is encoded into its latent representation $z_0 = \mathcal{E}(I)$ using the VQGAN encoder $\mathcal{E}$. For a chosen fixed timestep $t$, the corresponding noisy latent $z_t$ is computed via Eq. 1. This $z_t$ is then processed by the denoising U-Net $\epsilon_\theta(\cdot)$. However, as the network $\epsilon_\theta$ is trained to predict noise

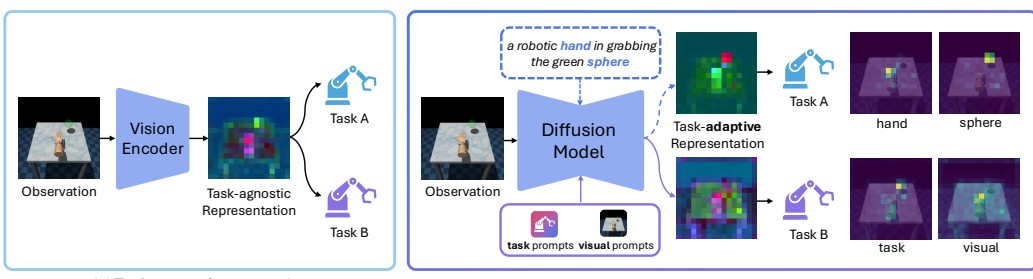

Figure 2: **Motivation.** We aim to overcome the limitations of existing **task-agnostic** approach (a) with frozen pre-trained visual representations (Parisi et al., 2022), by leveraging **conditions** in diffusion models for robotic control tasks in a **task-adaptive** approach (b). In this regard, we explore text conditions(§ 4.1), more advanced methods(§ 4.2,§ 5) as conditions.

as shown in Eq. 2, we instead extract intermediate feature maps from within the U-Net (Meng et al., 2024). We denote the set of extracted intermediate features as $f$, and denote $f = \epsilon_\theta(z_t, t; \mathcal{C})$ to be the output of $\epsilon_\theta$ for simplicity, and primarily consider features from the earlier blocks of the U-Net.

# 4 MOTIVATION

In this work, we explore conditional diffusion models to generate visual representations for robotic control, aiming to overcome the limitations of task-agnostic approaches. While pre-trained visual representations have been paramount to advancements in control, the standard approach of deploying the same frozen representation across various tasks often fails to adapt to their specific requirements, causing performance to fluctuate significantly (Majumdar et al., 2023). We aim to address this limitation by leveraging text-to-image diffusion models, which have successfully handled diverse visual tasks in a **task-adaptive** manner using well-designed textual prompts as conditions. Our goal is therefore to explore effective ways to condition diffusion models for control, as illustrated in Fig. 2.

However, we find that **text conditions are ineffective in robotic control environments** (§ 4.1), as using captions generated from vision-language models yields insignificant gains, or even degrades performance. An in-depth inspection of the cross-attention maps reveals the underlying reason for this failure - in tasks where performance degrades, the diffusion model struggles to correctly associate words with their corresponding image regions. This underscores the need for alternatives to text descriptions and for careful consideration when devising conditions specifically for robotic control.

Consequently, we discuss what do we need for effectively conditioning diffusion models in robotic control (§ 4.2). By their nature, control tasks involve video frames with fine-grained movements of agents and objects. Relying solely on textual conditions would necessitate generating a highly detailed, frame-by-frame description of the specific agent parts relevant to the current action—a challenging and often impractical task. Therefore, we posit that we should **incorporate visual information** for effective conditions to capture the fine-grained details of each frame.

## 4.1 EXPLORING TEXTUAL CONDITIONS FOR ROBOTIC CONTROL

To obtain textual descriptions of control environments, we devise a baseline by prompting a state-of-the-art vision-language model, Gemini 2.5 (Comanici et al., 2025), to generate descriptions of these tasks. The full text descriptions are provided in the appendix. For our analysis, we compare the null ($\varnothing$) condition—implemented as an empty string with only `<eos>` and `<bos>` tokens—and the text condition in downstream control tasks. However, as observed in Fig. 3(a), the results are mixed: while text conditions benefit some tasks (*e.g.*, Button-press, Reacher), they degrade performance in others (*e.g.*, Cheetah-run).

To take a deeper look, in Fig. 3(b), we visualize the cross-attention maps for *Button-press*, a task where text conditions show noticeable gains. For words such as *press* or *button*, the cross-attention maps are well-associated with the relevant regions within the image. These results are similar to what is expected from text conditions in other visual perception tasks like semantic segmentation (Zhao et al., 2023), which verifies the potential of using conditions in control tasks.

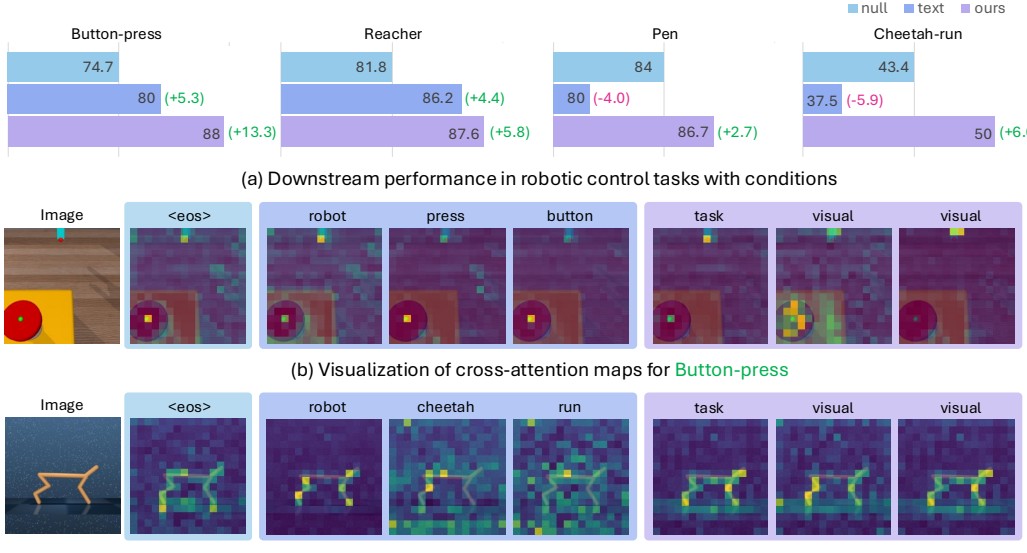

(a) Downstream performance in robotic control tasks with conditions

(b) Visualization of cross-attention maps for Button-press

(c) Visualization of cross-attention maps for Cheetah-run

Figure 3: **Case study.** **(a)** We find that text conditions can be disadvantageous in some control tasks. **(b)** For *Button-press*, the cross-attention maps (e.g., for *button*, *press*) are well-grounded to relevant image regions. **(c)** In contrast, for *Cheetah-run*, the attention maps (e.g., for *cheetah*, *run*) are noisy, which presumably leads to a decline in performance. Nonetheless, our approach of using task and visual tokens (§ 5) achieves consistent gains across all tasks, with its cross-attention maps capturing diverse regions of the image relevant to the downstream task.

However, in Fig. 3(c), we observe the opposite for *Cheetah-run*, where words like *cheetah* or *run* show noisy cross-attention maps. The <eos> token of the null condition is already roughly grounded to the salient object, the agent in this case, which explains how a sub-optimal text condition can degrade performance to be even worse than the null condition. We primarily attribute the failure of text conditions, despite being generated from a state-of-the-art vision-language model, to the domain gap between real-world images and simulated control environments. This finding highlights the need for careful consideration when devising conditions in robotic control and motivates the exploration of alternatives to text descriptions for representing the task.

## 4.2 WHAT DO WE NEED AS CONDITIONS IN ROBOTIC CONTROL?

In order to devise effective conditions, we discuss the characteristics of robotic control tasks and contrast with other vision-based tasks, such as semantic segmentation. A primary distinction is that control tasks operate on video streams rather than static images. Consequently, a logical approach would be to generate a unique condition for each frame (Hong et al., 2024), allowing the representation to adapt to the changing visual state of the environment. For instance, instructing an agent to walk requires a sequence of distinct commands (*e.g.*, move the left foot, then the right). Similarly, an effective condition should vary across frames to guide such dynamic behaviors. However, generating high-quality text descriptions on a frame-by-frame basis would not only be challenging but would also inherit the same grounding limitations discussed previously.

In this regard, we hypothesize that to account for this dynamic adaptability, conditions should incorporate **visual** information from each frame. While diffusion models like Stable Diffusion are typically trained on text, several approaches exist for incorporating visual information, either by introducing features from external vision encoders (Li et al., 2024; Patni et al., 2024) or by optimizing specialized text tokens to represent visual concepts (Kondapaneni et al., 2024; Kim et al., 2025). These existing methods, however, tend to embed the global representation into the condition, or require additional optimization steps to acquire specialized tokens. Since our goal is to enable the recognition of fine-grained regions within each frame, we consider that adopting global representations and extra optimization steps should be avoided to facilitate effective frame-wise conditioning.

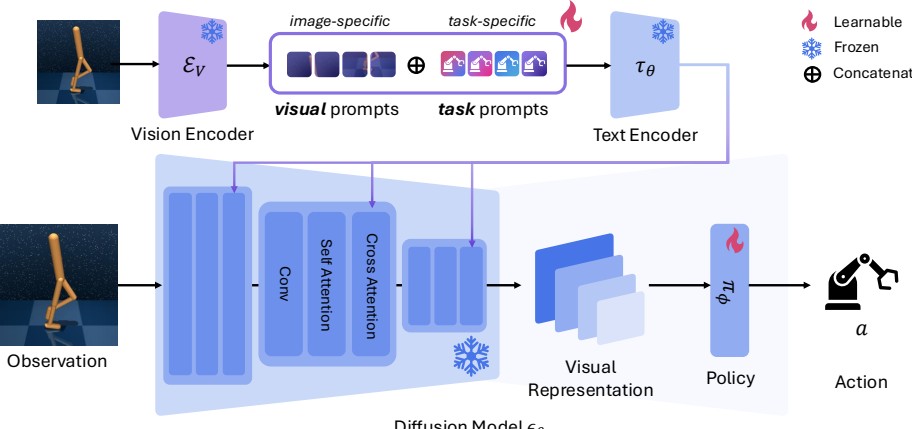

Figure 4: **Proposed framework.** We propose CoRoCo, a framework for learning **task** and **visual** prompts to condition diffusion models in robotic control. Specifically, we utilize the features from the downsampling blocks and the bottleneck block of Stable Diffusion (Rombach et al., 2022) to extract visual representations conditioned on our input, which are then fed to the policy network for predicting the action.

## 5 CoRoCo: Conditioning diffusion models for robotic control

Based on our observations, we present CoRoCo, a simple yet effective approach that learns prompts to condition diffusion models for control. We devise our condition adapt to the control environment to prevent erroneous grounding, while simultaneously incorporating visual information to capture dynamic details. To achieve this, we introduce learnable **task** and **visual** prompts, which are described in detail below.

**Task prompts.** Recalling that text conditions show potential when well-grounded to task-relevant regions, we design our task prompt to capture objects or areas that are critical to solving the downstream task. To achieve this, we implement task prompts as learnable parameters that are shared across all observations during training. We find that this allows the task prompts to implicitly learn to focus on relevant regions, as shown in Fig. 3(b,c), where the cross-attention maps simultaneously highlight both the button and the robot arm in *Button-press* and the agent in *Cheetah-run*.

**Visual prompts.** Furthermore, to incorporate visual information into the conditions, we adopt a vision encoder $\mathcal{E}_V$ to leverage its visual representation as prompts. Specifically, we utilize the dense visual representations from $\mathcal{E}_V$, rather than global representations, and project them through a small convolutional layer to complement the task prompts. This focus on dense features provides the fine-grained, localized information necessary for control tasks. As visualized in Fig. 3(b,c), the resulting attention from the visual prompts highlights various regions in detail, such as distinguishing between the front and back legs of the agent in *Cheetah-run*.

**Policy learning objective.** We learn the prompts by directly optimizing for the behavior cloning objective in downstream policy learning, as presented in Fig. 4. Let $\pi_\phi(\cdot)$ be the policy network with parameters $\phi$ that takes the visual state representations derived by the diffusion model and outputs actions. Given sequences of $T_o$ observations $\{I_o^i\}_{o=0}^{T_o}$ and actions $\{a_o^i\}_{o=0}^{T_o}$ from the $i$-th trajectory, we predict each action and train both the policy network $\pi_\phi(\cdot)$, task prompts $p_t$ and visual prompts $p_t$ by the behavior cloning loss:

$$\mathcal{L}_{\text{BC}(\phi,\mathbf{p})} = \sum_{i=1}^{N} \sum_{o} ||\pi_\phi(\epsilon_\theta(z_t, t; \mathcal{C}^*)) - a_o^i||, \tag{3}$$

where $z_t = \sqrt{\bar{\alpha}_t}\mathcal{E}(I_o^i) + \sqrt{1 - \bar{\alpha}_t}\epsilon$, and condition $\mathcal{C}^* = \tau_\theta(p_t; p_v)$ is derived from the text encoder $\tau_\theta$ with task prompt $p_t$ and visual prompt $p_v$ as the input. We find that $p_v$ and $p_t$ can be both learned with the behavior cloning loss in downstream policy learning.

Table 1: **Experimental results on vision-based robot policy learning on DeepMind Control**. The performance of imitation learning agents on DeepMind Control (Tassa et al., 2018) is reported. We report the normalized score averaged over three seeds with its standard deviation.

| Methods | Backbone | Walker-stand | Walker-walk | Reacher-easy | Cheetah-run | Finger-spin | Mean |
|---|---|---|---|---|---|---|---|
| CLIP | ViT-L/16 | $87.3 \pm 2.4$ | $58.3 \pm 4.4$ | $54.5 \pm 4.6$ | $29.9 \pm 5.6$ | $67.5 \pm 2.1$ | 59.5 |
| VC-1 | ViT-L/16 | $86.1 \pm 0.9$ | $54.3 \pm 6.6$ | $18.3 \pm 2.4$ | $40.9 \pm 2.7$ | $65.7 \pm 1.1$ | 53.1 |
| SCR | SD 1.5 | $85.5 \pm 2.6$ | $64.3 \pm 3.5$ | $81.8 \pm 9.9$ | $43.4 \pm 6.4$ | $66.6 \pm 2.7$ | 68.3 |
| Text (Simple) | SD 1.5 | $87.6 \pm 4.6$ | $67.9 \pm 4.6$ | $84.3 \pm 4.6$ | $38.8 \pm 5.9$ | $66.7 \pm 0.2$ | 69.1 |
| Text (Caption) | SD 1.5 | $87.2 \pm 4.5$ | $68.3 \pm 5.9$ | $86.2 \pm 1.9$ | $37.5 \pm 2.6$ | $65.1 \pm 1.8$ | 68.9 |
| CoOp | SD 1.5 | $87.2 \pm 2.2$ | $67.8 \pm 6.4$ | $87.1 \pm 5.9$ | $45.0 \pm 6.4$ | $65.9 \pm 1.0$ | 70.6 |
| TADP | SD 1.5 | $89.0 \pm 2.9$ | $69.9 \pm 7.9$ | $86.6 \pm 5.6$ | $41.1 \pm 3.9$ | $66.9 \pm 0.2$ | 70.7 |
| **CoRoCo (Ours)** | SD 1.5 | $\mathbf{89.1} \pm 1.8$ | $\mathbf{76.9} \pm 4.0$ | $\mathbf{87.6} \pm 2.9$ | $\mathbf{50.0} \pm 8.4$ | $\mathbf{68.0} \pm 1.0$ | **74.3** |

Table 2: **Experimental results on vision-based robot policy learning on MetaWorld**. The performance of imitation learning agents on MetaWorld (Yu et al., 2020) is reported. We report the success rates (%) averaged over three seeds with their standard deviation.

| Methods | Backbone | Assembly | Bin-picking | Button-press | Drawer-open | Hammer | Mean |
|---|---|---|---|---|---|---|---|
| CLIP | ViT-L/16 | $85.3 \pm 12.2$ | $69.3 \pm 8.3$ | $60.0 \pm 13.9$ | $\mathbf{100.0} \pm 0.0$ | $92.0 \pm 8.0$ | 81.3 |
| VC-1 | ViT-L/16 | $93.3 \pm 6.1$ | $61.3 \pm 12.2$ | $73.3 \pm 8.3$ | $\mathbf{100.0} \pm 0.0$ | $93.3 \pm 6.1$ | 84.2 |
| SCR | SD 1.5 | $92.0 \pm 6.9$ | $86.7 \pm 4.6$ | $74.7 \pm 12.9$ | $\mathbf{100.0} \pm 0.0$ | $98.7 \pm 2.3$ | 90.4 |
| Text (Simple) | SD 1.5 | $97.3 \pm 2.3$ | $85.3 \pm 2,3$ | $78.7 \pm 2,3$ | $\mathbf{100.0} \pm 0.0$ | $96.0 \pm 6.9$ | 91.5 |
| Text (Caption) | SD 1.5 | $96.0 \pm 4.0$ | $88.0 \pm 6.9$ | $80.0 \pm 8.3$ | $\mathbf{100.0} \pm 0.0$ | $98.7 \pm 2.3$ | 92.5 |
| CoOp | SD 1.5 | $96.0 \pm 4.0$ | $89.3 \pm 2,3$ | $81.3 \pm 6.1$ | $\mathbf{100.0} \pm 0.0$ | $96.0 \pm 6.9$ | 92.5 |
| TADP | SD 1.5 | $96.0 \pm 4.0$ | $\mathbf{90.7} \pm 4.6$ | $80.0 \pm 10.6$ | $\mathbf{100.0} \pm 0.0$ | $96.0 \pm 4.0$ | 93.1 |
| **CoRoCo (Ours)** | SD 1.5 | $\mathbf{98.7} \pm 2.3$ | $\mathbf{90.7} \pm 4.6$ | $\mathbf{88.0} \pm 6.9$ | $\mathbf{100.0} \pm 0.0$ | $\mathbf{98.7} \pm 2.3$ | **95.2** |

## 6 EXPERIMENTS

In this section, we establish the details for the evaluation (§ 6.1) and the implementation (§ 6.2) of our method, and present extensive experimental results (§ 6.4) and analyses (§ 6.5). We also provide further additional details ((§ C) and analyses (§ B) in the appendix.

### 6.1 EVALUATION SUITES

We conduct experiments on three widely-used vision-based robot learning benchmarks with the total of 12 tasks following VC-1 (Majumdar et al., 2023), as shown in Fig. 5.

**DeepMind Control (DMC)** (Tassa et al., 2018) is a set of continuous control tasks with simulated robots. We use five imitation learning cases: *Walker-stand*, *Walker-walk*, *Reacher-easy*, *Cheetah-run*, and *Finger-spin*. We report the normalized scores for all tasks.

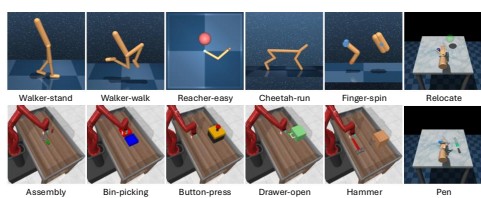

Figure 5: **Visualization of evaluation tasks.**

**MetaWorld** (Yu et al., 2020) is a suite of simulated robotic manipulation tasks with a Sawyer robot arm. We focus on a subset of five representative tasks: *Assembly*, *Bin-picking*, *Button-press-topdown*, *Drawer-open*, and *Hammer*. We measure the best success rates among the online evaluation trials.

**Adroit** (Rajeswaran et al., 2018) is an imitation learning benchmark in a simulated environment, consisting of dexterous manipulation tasks that require an agent to control a 28-DoF anthropomorphic hand. Our study mainly focuses on *Relocate* and *Pen*, and measure the best success rates among the online evaluation trials.

### 6.2 IMPLEMENTATION DETAILS

**Diffusion model and conditions.** We employ Stable Diffusion v1.5 (Rombach et al., 2022) as the diffusion model. For extracting visual representation from observations, we leverage the features from the downsampling blocks and the bottleneck block in the diffusion U-Net and forward through a compression layer (Yadav et al., 2023). We set the timestep $t = 0$, the length of task tokens $l_t = 4$,

Table 3: **Experimental results on vision-based robot policy learning on Adroit**. The performance of imitation learning agents on Adroit (Rajeswaran et al., 2018) is reported. We report the success rates (%) averaged over three seeds with their standard deviation.

| Methods | Backbone | Pen | Relocate | Mean |
|---|---|---|---|---|
| CLIP | ViT-L/16 | $58.7 \pm 2.3$ | $\mathbf{44.0} \pm 4.0$ | 51.4 |
| VC-1 | ViT-L/16 | $65.3 \pm 16.7$ | $29.3 \pm 8.3$ | 47.3 |
| SCR$^\dagger$ | SD 1.5 | $84.0 \pm 4.0$ | $32.0 \pm 4.0$ | 58.0 |
| Text (Simple) | | $80.0 \pm 6.9$ | $34.7 \pm 6.1$ | 57.3 |
| Text (Caption) | | $80.0 \pm 4.0$ | $34.7 \pm 4.6$ | 57.3 |
| CoOp | SD 1.5 | $82.7 \pm 6.1$ | $33.3 \pm 6.1$ | 58.0 |
| TADP | | $81.3 \pm 6.1$ | $33.3 \pm 8.3$ | 57.3 |
| **CoRoCo (Ours)** | | $\mathbf{86.7} \pm 2.3$ | $\mathbf{44.0} \pm 4.0$ | **65.3** |

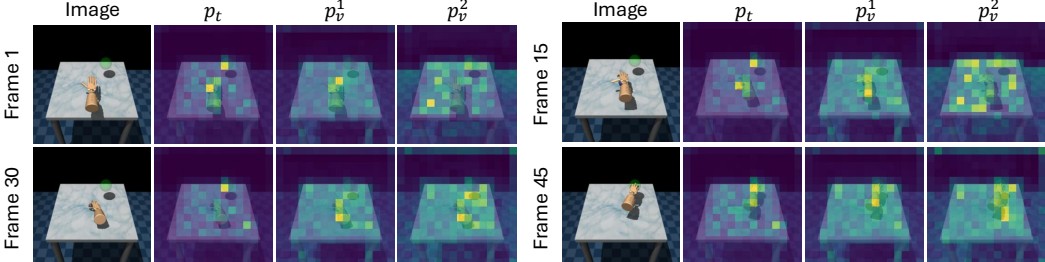

Figure 6: **Cross-attention visualization for task/visual prompts.** We visualize the cross-attention maps for task prompt $p_t$ and visual prompts $p_v^1$ and $p_v^2$ in *Relocate* task across frames from Adroit.

and the length of visual tokens $l_v = 16$, where all learnable parameters are randomly initialized. For $\mathcal{E}_V$, we employ pre-trained DINOv2 (Oquab et al., 2023). Further implementation details are presented in the appendix.

**Vision-based robot policy learning.** We consider two, five, and five demonstrations from Adroit, DeepMind Control (DMC), and MetaWorld, respectively, where proprioceptive data is utilized except for the DMC benchmark. We mainly follow the experimental setups in VC-1 (Majumdar et al., 2023) except that we employ a compression layer for all baselines for fair comparison. For each task, we train the agent for 100 epochs, with a periodic online evaluation in the simulated environment every 10 epochs.

## 6.3 BASELINES

We consider three baselines: CLIP (Radford et al., 2021), VC-1 (Majumdar et al., 2023), and SCR (Gupta et al., 2024), as *task-agnostic* baselines, which follow the standard frozen pre-trained visual representation approach. In addition, we provide four *task-adaptive* baselines: Text$_{\text{simple}}$, Text$_{\text{caption}}$, CoOp (Zhou et al., 2022), TADP (Kondapaneni et al., 2024). We provide details for the baselines in the appendix.

## 6.4 MAIN RESULTS

**Quantitative results.** We report experimental results from DMC, MetaWorld, and Adroit in Table 1, Table 2, and Table 3, respectively. Among the task-agnostic baselines, while SCR performs best overall, we observe that VC-1 and CLIP outperform it in certain tasks. This highlights a fundamental limitation of such approaches: due to their task-agnostic nature, no single representation is guaranteed to excel across all tasks. In contrast, across all 12 tasks in the 3 evaluation suites, CoRoCo establishes the new state-of-the-art, outperforming all baselines by a significant margin.

Furthermore, we observe that more advanced task-adaptive baselines, CoOp and TADP, generally outperform text conditions, which aligns with our analysis and confirms our hypothesis that incorporating visual information is critical. Nonetheless, since CoOp and TADP were originally designed for tasks like image classification and semantic segmentation, their effectiveness in robotic control is limited, as shown by their minimal gains on DMC and Adroit. In contrast, our method show solid improvements across all tasks.

Table 4: **Components analysis**. To ablate the design choices for learning conditions, we conduct component analysis on task prompt $p_t$ and visual prompt $p_v$. The performance of imitation learning agents on DeepMind Control (Tassa et al., 2018) is reported. We report the normalized score averaged over three seeds with its standard deviation.

| Components | | DeepMind Control | | | | | |
|---|---|---|---|---|---|---|---|
| $p_t$ | $p_v$ | Walker-stand | Walker-walk | Reacher-easy | Cheetah-run | Finger-spin | Mean |
| | | $85.5 \pm 2.6$ | $64.3 \pm 3.5$ | $81.8 \pm 1.7$ | $43.4 \pm 4.4$ | $66.6 \pm 2.7$ | 68.3 |
| ✓ | | $83.6 \pm 3.2$ | $71.4 \pm 3.5$ | $86.7 \pm 6.6$ | $38.9 \pm 10.1$ | $\mathbf{68.2} \pm 1.2$ | 69.8 |
| | ✓ | $85.9 \pm 2.7$ | $71.1 \pm 2.3$ | $87.3 \pm 5.5$ | $42.0 \pm 10.4$ | $66.1 \pm 1.0$ | 70.5 |
| ✓ | ✓ | $\mathbf{89.1} \pm 2.3$ | $\mathbf{76.9} \pm 4.0$ | $\mathbf{87.6} \pm 2.9$ | $\mathbf{50.0} \pm 8.4$ | $68.0 \pm 1.0$ | $\mathbf{74.3}$ |

Table 5: **Ablation study on layer selection**. We evaluate individual layers of the diffusion U-Net by reporting layer-wise performance. The performance of imitation learning agents on DeepMind Control (Tassa et al., 2018) is reported. We report the normalized score averaged over three seeds with its standard deviation.

| Layer | DeepMind Control | | | | | |
|---|---|---|---|---|---|---|
| | Walker-stand | Walker-walk | Reacher-easy | Cheetah-run | Finger-spin | Mean |
| down_1 | $86.3 \pm 2.1$ | $65.5 \pm 1.1$ | $82.1 \pm 3.7$ | $40.8 \pm 1.1$ | $67.6 \pm 0.3$ | 68.4 |
| down_2 | $\mathbf{89.3} \pm 1.2$ | $68.3 \pm 2.7$ | $70.0 \pm 18.8$ | $31.2 \pm 2.6$ | $67.0 \pm 1.0$ | 65.1 |
| down_3 | $86.2 \pm 4.3$ | $73.3 \pm 3.9$ | $75.3 \pm 8.1$ | $36.0 \pm 4.8$ | $67.0 \pm 0.5$ | 67.5 |
| mid | $88.3 \pm 4.9$ | $70.4 \pm 1.3$ | $62.3 \pm 1.1$ | $35.0 \pm 4.7$ | $67.2 \pm 0.6$ | 64.6 |
| up_0 | $82.8 \pm 2.6$ | $71.7 \pm 5.9$ | $45.3 \pm 4.0$ | $28.5 \pm 1.8$ | $67.2 \pm 0.6$ | 59.0 |
| up_1 | $79.5 \pm 4.5$ | $60.3 \pm 16.1$ | $55.9 \pm 5.2$ | $39.9 \pm 7.0$ | $66.4 \pm 0.4$ | 60.4 |
| up_2 | $70.4 \pm 4.5$ | $39.1 \pm 3.3$ | $41.0 \pm 7.0$ | $30.9 \pm 3.1$ | $67.7 \pm 1.0$ | 49.7 |
| down_1-3, mid | $89.1 \pm 1.8$ | $\mathbf{76.9} \pm 4.0$ | $\mathbf{87.6} \pm 2.9$ | $\mathbf{50.0} \pm 8.4$ | $\mathbf{68.0} \pm 1.0$ | $\mathbf{74.3}$ |

## 6.5 ANALYSIS

**Visualization of task and visual prompts.** In Fig. 6, we visualize the cross-attention maps for our task prompt $p_t$ and visual prompts, $p_v^1$ and $p_v^2$, on *Relocate*. In this task, a robot hand first picks up a blue ball from a table (Frames 1-30) and then moves it to the location of a green sphere (Frames 30-45). As discussed in § 5, we observe that the task prompt consistently captures regions relevant to the overall goal, namely the robot hand and the target green sphere. Conversely, the visual prompts exhibit more dynamic behaviors. While $p_v^1$ tends to focus on the hand, $p_v^2$ interestingly attends to the table as the hand moves downward to pick up the ball, then shifts its focus to the hand as it lifts off and moves toward the target, suggesting that it has learned to capture task-relevant movements.

**Ablation study on each component.** In Table 4, we conduct component analysis by ablating task prompt $p_t$ and visual prompt $p_v$ respectively. Notably, we observe that when employed individually, task and visual prompts can show divergent behavior across different tasks. This could suggest that each tasks focus in different aspects of the scene, such as *Reacher-easy* focusing more in visual details as it benefits more from visual prompts compared to text prompts. Nonetheless, when fully incorporating both task and image prompts, we show consistent gains across all tasks.

**Ablation study on layer selection.** In Table 5, we provide a layer-wise evaluation of the diffusion model. In practice, we leverage multi-layer features by concatenating the best-performing layers (down_1, down_2, down_3, mid), which yields the best overall performance. This layer selection coincides with SCR (Gupta et al., 2024), and also share findings with Parisi et al. (2022), where they find that representations from early layers of vision encoders perform better in robotic control tasks.

## 7 CONCLUSION

In this work, we introduced CoRoCo, a framework for bridging text-to-image diffusion models for robotic control to generate task-adaptive visual representations. We identified the limitations of conventional text prompts in control settings, and we proposed a simple yet effective method using learnable task and visual prompts. By training these prompts with the behavior cloning objective, CoRoCo achieves state-of-the-art performance, highlighting the importance of task-adaptive representations and the vast potential of properly conditioned diffusion models for robotic control.

**Reproducibility Statement** To ensure reproducibility, we have provided the implementation details in Sec. 6.2, and further specify details in Sec. C in the appendix. In Sec. 6.2, we have provided the hyperparameter setups for our method and the details of the pre-trained vision encoder and the diffusion model used in the method. In Sec. C, we provide details on the text conditions used in the baselines, and also provide details on the compression layer (Yadav et al., 2023), including its pseudo-code in Alg. 1.

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

## APPENDIX

## A  FURTHER ABLATION STUDIES

### A.1  ABLATION ON DIFFUSION TIMESTEPS

Table 6: **Ablation study on timestep selection**. To ablate the choice for timestep $t$, we provide results with $t = 100$ and $t = 200$. The performance of imitation learning agents on DeepMind Control (Tassa et al., 2018) is reported. We report the normalized score averaged over three seeds with its standard deviation.

| Timestep | DeepMind Control | | | | | |
| --- | --- | --- | --- | --- | --- | --- |
| | Walker-stand | Walker-walk | Reacher-easy | Cheetah-run | Finger-spin | Mean |
| 200 | **92.2** $\pm$ 1.6 | 78.6 $\pm$ 2.2 | **85.4** $\pm$ 8.3 | 24.7 $\pm$ 4.5 | 66.5 $\pm$ 3.2 | 69.4 |
| 100 | 88.3 $\pm$ 4.7 | 72.6 $\pm$ 4.3 | 79.4 $\pm$ 6.8 | 36.1 $\pm$ 6.2 | 66.2 $\pm$ 3.5 | 68.5 |
| **0** (Default) | 88.8 $\pm$ 2.3 | 76.9 $\pm$ 4.0 | 71.9 $\pm$ 5.6 | **48.2** $\pm$ 11.3 | **68.0** $\pm$ 1.0 | **70.8** |

To ablate the effects of the diffusion timestep $t$, in Table 6, we additionally provide results with $t = 100$ and $t = 200$. Although some tasks (*e.g. Reacher-easy*) benefit from $t = 100$ or $t = 200$, performance on other tasks such as *Cheetah-run* degrades significantly, lowering the overall score. Therefore, we choose $t = 0$, which achieves the best overall performance.

## B    FURTHER ANALYSIS

### B.1    ANALYSIS ON THE NULL CONDITION

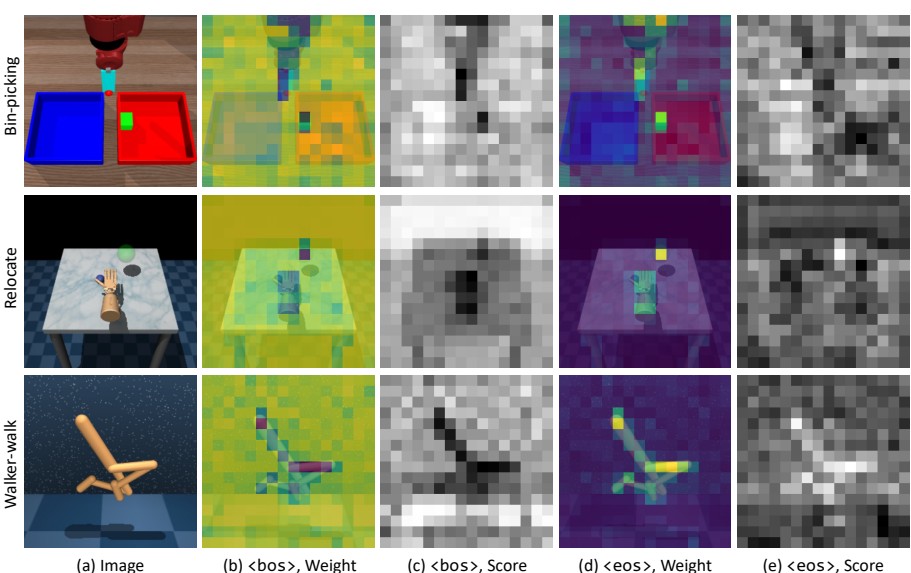

|  (a) Image | (b) `<bos>`, Weight | (c) `<bos>`, Score | (d) `<eos>`, Weight | (e) `<eos>`, Score |

Figure 7: **Visualization of normalized attention weights and raw attention scores for `<bos>` and `<eos>` tokens.** We compare the visualization of the normalized attention weights obtained after the softmax operation and the raw attention scores obtained before the softmax operation from the cross-attention layers to further analyze the properties of `<bos>` and `<eos>` tokens.

Figure 7 illustrates the attention behavior of `<bos>` and `<eos>` tokens by visualizing their normalized cross-attention maps (b,d) and raw attention scores (c,e). We observe that the `<bos>` token consistently attends to background regions, whereas the `<eos>` is less reliable at focusing on salient objects (*e.g.* the robot hand in *Relocate*). We attribute the background affinity of `<bos>` to the typical structure of text prompts, which primarily describe foreground subjects. Moreover, since Stable Diffusion employs a causal text encoder for both conditional prompts and the null condition ∅ in unconditional generation, this background-attending behavior is also transferred to unconditional scenarios.

### B.2    EFFICIENCY COMPARISON

In Table 7, we report the number of parameters, number of learnable parameters, and latency for each modules for VC-1 (Majumdar et al., 2023), SCR (Gupta et al., 2024) and our proposed method. For VC-1 and SCR, we use ViT-L/16, which was also used for comparison in the main paper. Notably, the layer selection allows us to drop the "up" blocks, which removes around 500M parameters from the denoising U-Net. This allows the U-Net to have similar parameter count to VC-1 encoder, which is used in various robotic manipulation tasks. Furthermore, most of the parameters added to our method are

Table 7: **Efficiency comparison**. We report the total number of parameters (#Params), the number of learnable parameters (#Learnable) and latency for VC-1, SCR, and ours.

| Method | #Params | #Learnable | Time |
|--------|---------|------------|------|
| VC-1   | 303.3M  | 0          | 11ms |
| SCR    | 382.9M  | 0          | 26ms |
| Ours   | 480.1M  | 10.6M      | 48ms |

the frozen parameters from DINOv2, and the learnable parameters consist mostly of the additional projection layers for the visual prompts.

## C  FURTHER IMPLEMENTATION DETAILS

### C.1  FULL DESCRIPTION OF THE TEXT CONDITIONS

In Table 8, we provide the full descriptions used for $\text{Text}_{\text{simple}}$ and $\text{Text}_{\text{caption}}$, which are generated by Gemini 2.5 Pro (Comanici et al., 2025). For CoOp (Zhou et al., 2022), we use 4 learnable prefix tokens, such as "$[V^*][V^*][V^*][V^*]$ bin picking" for *Bin-picking*. For TADP, we add a style prefix "in a $[S^*]$ style", which results in "The Sawyer robot arm must carefully pick a specific target object out of the cluttered red bin and place it into the empty blue bin in a $[S^*]$ style." for *Bin-picking*.

Table 8: **Full text descriptions used in baselines**.

| Task | Method | Text |
|---|---|---|
| Assembly | $\text{Text}_{\text{simple}}$ | "assembly" |
| | $\text{Text}_{\text{caption}}$ | "The Sawyer robot arm must pick up the green block and precisely insert it into the center of the silver ring to complete the assembly." |
| Bin | $\text{Text}_{\text{simple}}$ | "bin picking" |
| | $\text{Text}_{\text{caption}}$ | "The Sawyer robot arm must carefully pick a specific target object out of the cluttered red bin and place it into the empty blue bin." |
| Button | $\text{Text}_{\text{simple}}$ | "button press" |
| | $\text{Text}_{\text{caption}}$ | "The Sawyer robot arm must reach out and accurately press the red button on top of the yellow control box." |
| Drawer | $\text{Text}_{\text{simple}}$ | "drawer open" |
| | $\text{Text}_{\text{caption}}$ | "The Sawyer robot arm must grasp the white handle and pull open the light green drawer." |
| Hammer | $\text{Text}_{\text{simple}}$ | "hammer" |
| | $\text{Text}_{\text{caption}}$ | "The Sawyer robot arm must pick up the red hammer and use it to strike the nail, driving it completely into the wooden block." |
| Pen | $\text{Text}_{\text{simple}}$ | "pen" |
| | $\text{Text}_{\text{caption}}$ | "A dexterous robotic hand must twirl a blue pen within its grasp to match the final orientation shown by the green target pen." |
| Relocate | $\text{Text}_{\text{simple}}$ | "relocate" |
| | $\text{Text}_{\text{caption}}$ | "A dexterous robotic hand is tasked with picking up the small blue ball and moving it to the location of the green target sphere." |
| Cheetah-run | $\text{Text}_{\text{simple}}$ | "cheetah run" |
| | $\text{Text}_{\text{caption}}$ | "A minimalist orange robot, shaped like a cheetah, runs across a reflective floor in a simulated environment." |
| Walker-walk | $\text{Text}_{\text{simple}}$ | "walker walk" |
| | $\text{Text}_{\text{caption}}$ | "A minimalist, orange bipedal robot takes a step across a reflective floor in a simulated environment." |
| Walker-stand | $\text{Text}_{\text{simple}}$ | "walker stand" |
| | $\text{Text}_{\text{caption}}$ | "A minimalist, orange bipedal robot stands upright on a reflective floor in a simulated environment." |
| Finger-spin | $\text{Text}_{\text{simple}}$ | "finger spin" |
| | $\text{Text}_{\text{caption}}$ | "A simple robotic finger strikes a floating, hot dog-shaped object to make it spin against a starry background." |
| Reacher | $\text{Text}_{\text{simple}}$ | "reacher" |
| | $\text{Text}_{\text{caption}}$ | "A simple robotic arm reaches for a red target ball on a checkered blue surface." |

### C.2  DETAILS OF THE BASELINES

**CLIP** (Radford et al., 2021) is a vision-language model pre-trained on large-scale image-text pairs through contrastive learning. CLIP has been widely used in various tasks, including navigation and manipulation tasks (Shridhar et al., 2022; Khandelwal et al., 2022).

**VC-1** (Majumdar et al., 2023) is a foundation model for various robotics tasks, spanning from manipulation to locomotion and navigation tasks. VC-1 trains with MAE objective on egocentric videos, as well as additional data including navigation and manipulation datasets.

**SCR** (Gupta et al., 2024) employs Stable Diffusion for various navigation and manipulation tasks. We consider SCR as a baseline using the null condition $\varnothing$, which is implemented as an empty string.

**Text(Simple/Caption)** is a task-adaptive baseline using text conditions, where Text (Simple) directly uses the task names as the condition, whereas Text (Caption) leverages descriptions generated from Gemini 2.5 (Comanici et al., 2025). Full text used for each tasks are presented in the appendix.

**CoOp** (Zhou et al., 2022) extends on $\text{Text}_{\text{simple}}$ by implementing learnable prefix tokens $V^*$. CoOp originally prompts CLIP with the format "$[V^*]$ *classname*" for image classification, which in our case, the task names used in $\text{Text}_{\text{simple}}$ are used as classnames.

**TADP** (Kondapaneni et al., 2024) extends on $\text{Text}_{\text{caption}}$, by adding a special token $S^*$ that encapsulates the visual style information optimized through Textual Inversion (Gal et al., 2022). Since the visual information is optimized into a single token $S^*$, we can consider TADP as a baseline with global visual information, and not in a frame-wise manner.

### C.3  IMPLEMENTATION DETAILS OF THE COMPRESSION LAYER

To provide further details of the compression layer (Yadav et al., 2023), we provide a PyTorch-style pseudo-code of the compression layer in Alg. 1. We follow previous works (Yadav et al., 2023; Gupta et al., 2024) for implementing a simple convolutional layer for the compression layer to obtain 1D state representations from 2D features. For all methods, `compress_dim` was set to 48. Note that

---

**Algorithm 1:** PyTorch-style pseudocode for the compression layer

---

```
class CompressionLayer(nn.Module):
  def __init__(self, hidden_dim, compress_dim):
    self.layers = nn.Sequential(
        nn.Conv2d(hidden_dim, compress_dim, kernel_size=3,
         padding=1),
        nn.BatchNorm2d(hidden_dim),
        nn.ReLU(inplace=True),
        nn.Flatten()
        )
  def forward(self, x):
    return self.layers(x)
```

---

the compression layer was also used for compared baselines including CLIP (Cherti et al., 2023) and VC-1 (Majumdar et al., 2023), which have been shown to yield better performance than using `<CLS>` tokens (Gupta et al., 2024).

## D    LARGE LANGUAGE MODEL USAGE

As we have discussed in Sec. 4.1, we prompt a state-of-the-art large vision-language model, Gemini 2.5 Pro (Comanici et al., 2025) to generate text descriptions for control tasks, which are also specified Sec. C.1.

Otherwise than generating text descriptions, for writing the manuscript, we disclose that LLMs were only used to polish the writing and was **not** used in any research purpose, such as research ideation or retrieval of related work.

## E    QUALITATIVE VISUALIZATION ON ROBOTIC CONTROL TASKS

We provide frame-wise comparison of our method, CLIP (Cherti et al., 2023), and VC-1 (Majumdar et al., 2023) for tasks from DMC (Tassa et al., 2018) in Fig. 9, MetaWorld (Yu et al., 2020) in Fig. 10, and Adroit (Rajeswaran et al., 2018) in Fig. 8. For each task, we report the normalized score for DMC and whether the task has succeeded or failed for MetaWorld and Adroit.

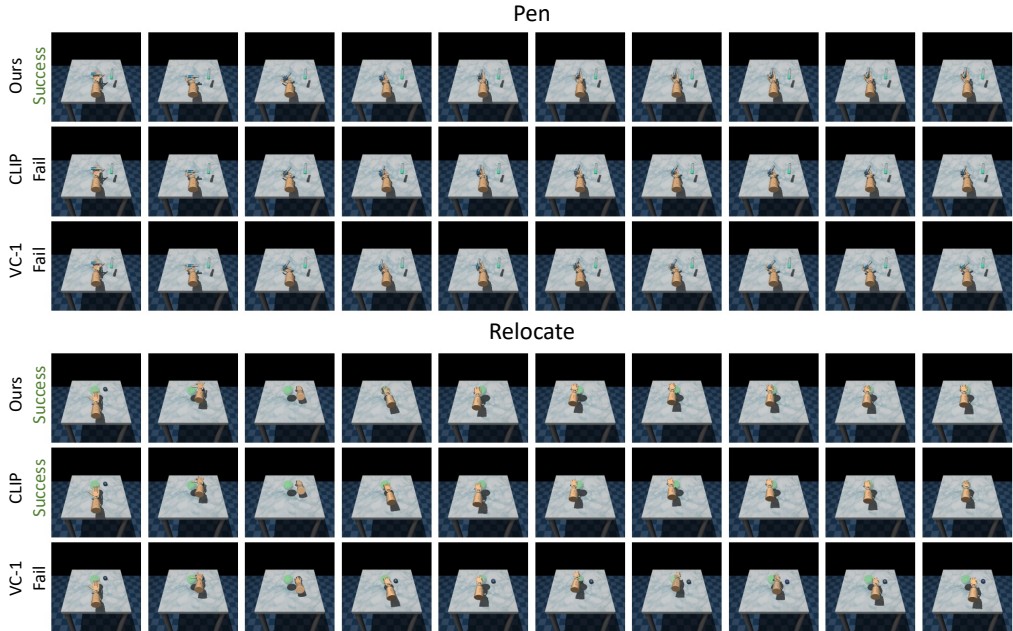

Figure 8: **Visualization of agents performing downstream tasks in Adroit (Rajeswaran et al., 2018).** We provide visual comparison of our method to CLIP (Cherti et al., 2023), and VC-1 (Majumdar et al., 2023) for two tasks from Adroit. We additionally report whether the task has succeeded or failed for each episode.

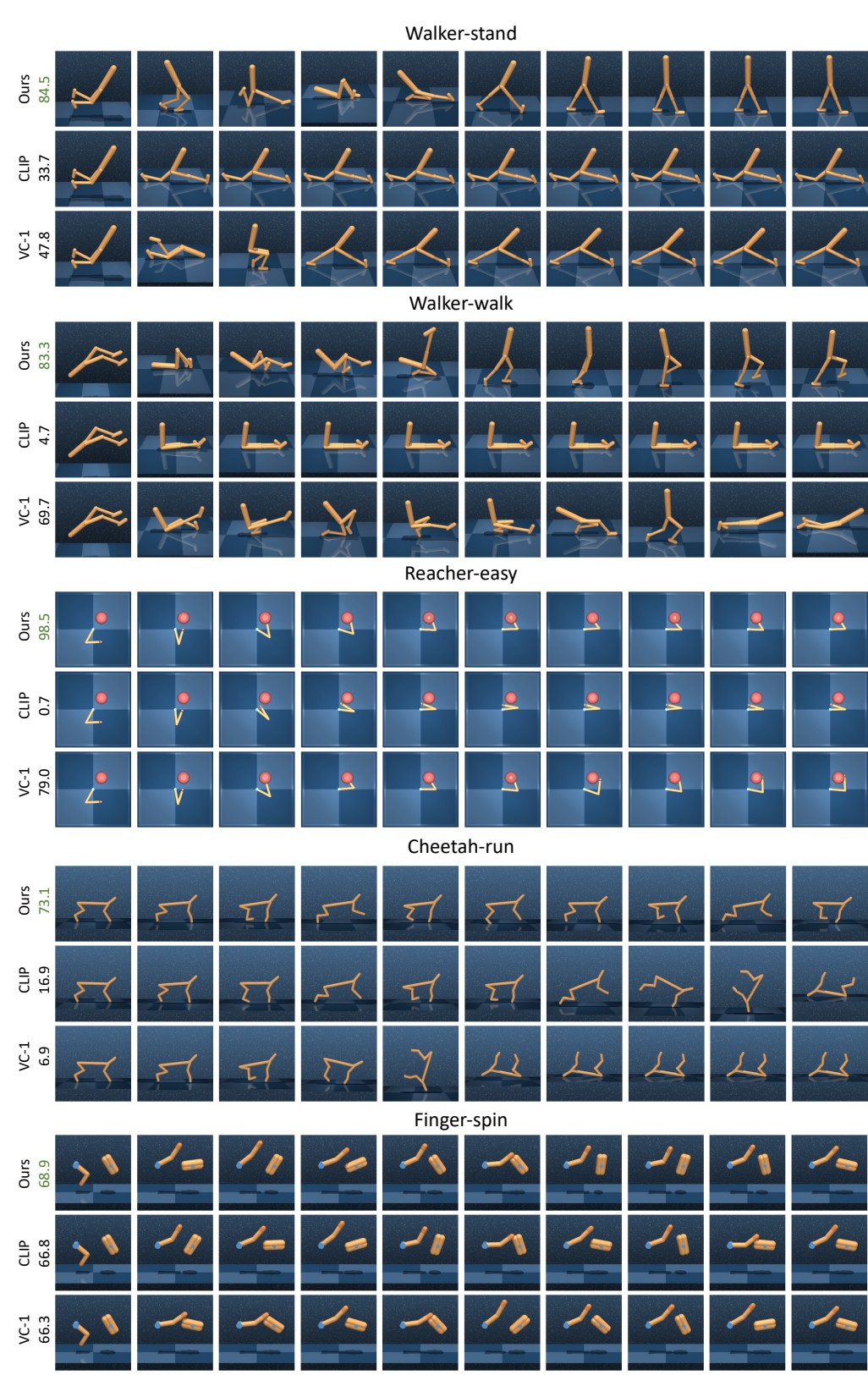

Figure 9: **Visualization of agents performing downstream tasks in DMC (Tassa et al., 2018).** We provide a visual comparison of our method to CLIP (Cherti et al., 2023), and VC-1 (Majumdar et al., 2023) for five tasks in DMC. We additionally report the normalized score for each episode.

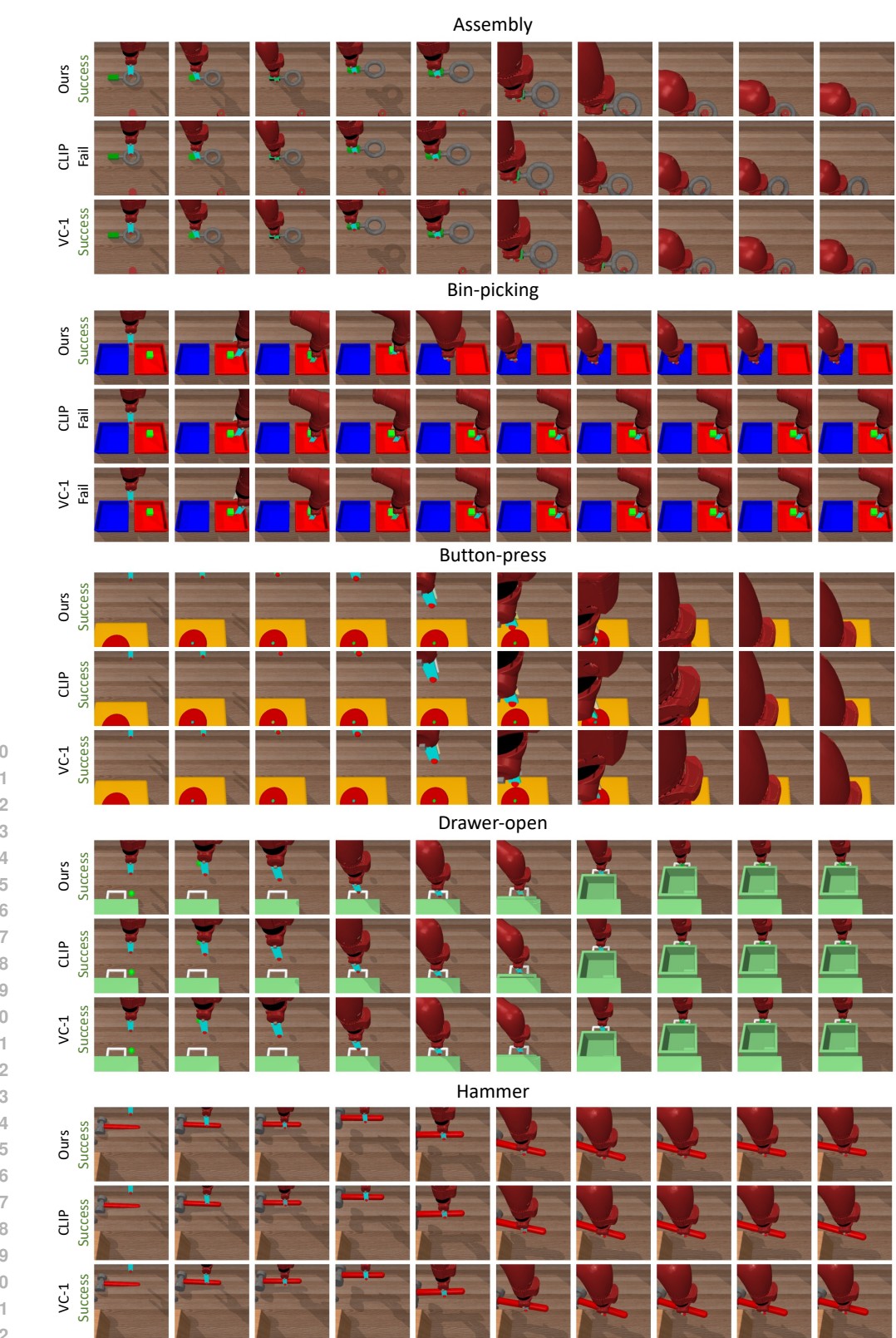

Figure 10: **Visualization of agents performing downstream tasks in MetaWorld (Yu et al., 2020).** We provide visual comparison of our method to CLIP (Cherti et al., 2023), and VC-1 (Majumdar et al., 2023) for five tasks in MetaWorld. We additionally report whether the task has succeeded or failed for each episode.

