# OpenReview forum: "Exploring Conditions for Diffusion models in Robotic Control"
_ICLR.cc/2026/Conference — ICLR 2026 Conference Withdrawn Submission_

### Official Review · Reviewer_bZ1v · 2025-10-26

**Soundness:** 3
**Presentation:** 4
**Contribution:** 3
**Rating:** 4
**Confidence:** 2

**Summary:**

CoRoCo leverages pre-trained text-to-image diffusion models to derive task-adaptive visual representations for robotic control without fine-tuning the backbone. Recognizing that naive textual conditioning underperforms due to domain gaps, CoRoCo introduces learnable task prompts and frame-specific visual prompts that capture dynamic, fine-grained control-relevant information, yielding state-of-the-art results across multiple robotic benchmarks.

**Strengths:**

This paper proposes a novel robot control strategy that uses learnable conditions to bridge the gap between vision pre-training and robotic environments. The motivation is sound, the method appears technically solid, and the experiments are fairly comprehensive. The paper is well written, the method is clearly presented, and the figures/tables are complete and easy to read.

**Weaknesses:**

The proposed approach adapts only a learnable condition token while freezing the remaining modules. This design is attractive because it substantially lowers training cost and retains the strengths of the pre-trained backbone. However, a key concern is whether such limited adaptation can guarantee downstream effectiveness and generalization. The current evaluations are concentrated on simulated locomotion and Meta-World manipulation—domains that are short-horizon and comparatively insensitive to language and rich visual semantics (indeed, many Meta-World tasks can be solved with images alone, without language). Demonstrating results on more complex, language- and vision-sensitive scenarios (e.g., longer-horizon, multi-object, compositional instructions, o.o.d. visuals) would better substantiate the generality of the approach.

**Questions:**

See above in weakness.

---

### Official Review · Reviewer_adaB · 2025-10-27

**Soundness:** 2
**Presentation:** 3
**Contribution:** 2
**Rating:** 4
**Confidence:** 5

**Summary:**

The paper presents CoRoCo, a diffusion-based framework that introduces learnable task and visual prompts to guide the generation of evolving actions for robotic control. After a dual-encoder architecture fuses textual and visual inputs into a shared SVD representation, a policy network maps the resulting visual features to the action space. CoRoCo is evaluated across multiple simulated environments against strong baselines, demonstrating superior success rates on various manipulation tasks.

**Strengths:**

- Using learnable tasks and visual prompts to replace the original diffusion model text condition, which highlights various regions in detail.
- The method builds upon an SVD backbone pre-trained on large-scale video datasets, while keeping the U-Net parameters frozen. This design preserves the model’s foundational generative ability but raises questions about the extent of adaptability to new domains.
- Visualization of experimental results and verification including different simulation environments.

**Weaknesses:**

- **Limited discussion with related work**
  The paper provides insufficient discussion and comparison with closely related approaches. For instance, *Vidman* [1] and *VPP* [2] similarly employ video diffusion models for single-step denoising as text–image encoders without freezing the visual representation. The primary difference between *CoRoCo* and these methods appears to be the modification of diffusion conditions through additional visual prompts.

- **Inadequate ablation analysis**
  The ablation experiments are weak and do not clearly demonstrate the contribution of freezing the SVD backbone to action learning. Intuitively, directly fine-tuning the SVD U-Net could also address the task-agnostic issue. Additional ablation studies are required to validate that the proposed learnable visual prompt mechanism outperforms full-parameter fine-tuning or LoRA alternatives.

- **Relatively weak experiment**
  The evaluation benchmarks (e.g., DMC and Metaworld) are not sufficiently challenging for visual–language–action tasks. For example, Metaworld tasks typically do not rely on textual inputs and can be solved effectively using visual information alone. Evaluations on more complex and realistic environments (e.g., *Robotwin* [3], *CALVIN* [4]) would strengthen the empirical claims.
Besides，when integrating textual information into policy learning, comparisons with recent Vision-Language-Action (VLA) frameworks—such as *π₀* [5]—would provide a more convincing validation of the proposed method’s advantage.

[1] Wen, Youpeng, et al. "Vidman: Exploiting implicit dynamics from video diffusion model for effective robot manipulation." Advances in Neural Information Processing Systems 37 (2024): 41051-41075.

[2] Hu, Yucheng, et al. "Video prediction policy: A generalist robot policy with predictive visual representations." arXiv preprint arXiv:2412.14803 (2024).

[3] Mu, Yao, et al. "Robotwin: Dual-arm robot benchmark with generative digital twins (early version)." European Conference on Computer Vision. Cham: Springer Nature Switzerland, 2024.

[4] Mees, Oier, et al. "Calvin: A benchmark for language-conditioned policy learning for long-horizon robot manipulation tasks." IEEE Robotics and Automation Letters 7.3 (2022): 7327-7334.

[5] Black, Kevin, et al. "$\pi_0 $: A Vision-Language-Action Flow Model for General Robot Control." arXiv preprint arXiv:2410.24164 (2024).

**Questions:**

1. Is it better to freeze SVD and use learnable visual prompts than to fine-tune U-Net in SVD directly? Please provide some relevant experiments.
2. Considering comparison with VPP and π₀, it is recommended to choose a benchmark that is more suitable for multimodal evaluation (such as Calvin)

**Details Of Ethics Concerns:**

Please see above weaknesses

---

### Official Review · Reviewer_n6on · 2025-11-02

**Soundness:** 2
**Presentation:** 3
**Contribution:** 1
**Rating:** 2
**Confidence:** 3

**Summary:**

The paper studies how to condition pre-trained text-to-image diffusion models so their internal features become task-adaptive visual representations for imitation/control, without fine-tuning the diffusion model itself. They show that off-the-shelf text prompts (captions) give inconsistent or negative gains in control because of grounding failures and the domain gap; they therefore propose CoRoCo: learnable task prompts (shared tokens) plus visual prompts (dense features from a frozen vision encoder) that together form the condition to a frozen Stable Diffusion U-Net; the U-Net’s intermediate features are passed to a policy and everything except the diffusion model (and mostly the vision encoder) is trained end-to-end with behavior cloning. Empirically CoRoCo improves across 12 tasks on DMC/MetaWorld/Adroit.

**Strengths:**

The presentation is clear and the paper presents a range of experimental results (visualization, ablations) to give the reader more insight.

**Weaknesses:**

1) My main issue is that I’m not convinced that this work has potential for impact in the area/problem being studied. The method is very similar to SCR which is compared to in the results section but the paper does not attribute to it all of the design choices related to feature extraction that are taken directly from that work (even the analysis and ablations present some very similar results for eg. impact of textual conditioning etc) - so it is unclear what the real contribution of this work is.

2) Architecture seems very arbitrary and the authors have not explained their choices. For eg. Why did they need dino to get visual conditioning tokens, shouldn’t the U-net itself have captured visual features (and only the task conditioning tokens should have been needed?)

3) The diffusion model architecture has also changed a lot since stable diffusion and the authors should have tried using a DiT [1] based architecture.

4) The experiments are on synthetic and saturated benchmarks where I‘m not sure it is very meaningful to show minor improvements. On Metaworld there is only 1 task (button press) where there is actually a meaningfully improvement. The approach proposed is just generally overkill on such simple tasks.

5) The proposed approach has so many more parameters, making it an unfair comparison to baselines. To show the need of foundation model backbones, the authors should show results across diverse and hard tasks (e.g. tasks that require zero/few-shot generalisation) to justify the spend on compute.

[1] Peebles, William, and Saining Xie. "Scalable diffusion models with transformers." Proceedings of the IEEE/CVF international conference on computer vision. 2023.

**Questions:**

* Why does the paper not report SOTA manipulation baselines (like R3M [1] and more that might have come afterwards).

[1] Nair, Suraj, et al. "R3m: A universal visual representation for robot manipulation." arXiv preprint arXiv:2203.12601 (2022).

---

### Note · Authors · 2025-11-12

**Comment:**

We appreciate the comments from the reviewers, and look forward to improve our paper based on the constructive feedback.

**Withdrawal Confirmation:**

I have read and agree with the venue's withdrawal policy on behalf of myself and my co-authors.